# Effect of COVID-19 on Internet Usage of People with Disabilities: A Secondary Data Analysis

**DOI:** 10.3390/ijerph19137813

**Published:** 2022-06-25

**Authors:** Eun-Young Park

**Affiliations:** Department of Secondary Special Education, Jeonju University, Jeonju 55069, Korea; eunyoung@jj.ac.kr; Tel.: +82-63-220-3186

**Keywords:** COVID-19 pandemic, internet use, non-face-to-face services, people with disabilities

## Abstract

The coronavirus disease (COVID-19) pandemic has led our society to lead a life different than before. People, including those with disabilities, have come to rely on information from the Internet. However, there is a lack of empirical studies on the impact of the pandemic on the Internet usage of people with disabilities. To address this gap, this study analyzed data from the 2020 Digital Divide Survey, which comprised data on 7000 non-disabled people and 2200 people with disabilities. This was used to compare the changes in usage of Internet services, and awareness and experience of COVID-19-related non-face-to-face services among non-disabled people and people with physical disabilities, brain lesions, visual impairments, hearing impairments, and language impairments. People with and without disabilities reported increased internet usage, but the increase was significantly higher in the non-disabled population than in people with disabilities (*p* < 0.05), except for people with language impairments. Specific changes to service usage, experience, and usefulness were different according to the type of disability. The non-disabled population showed a significantly greater increase in the use of social participation services than people with physical disabilities (*p* < 0.05). The results of this study suggest that digital services need to be developed flexibly to address the unique needs of people with different types of disabilities.

## 1. Introduction

As the coronavirus disease (COVID-19) spreads around the world, our society is experiencing a new daily life that is completely different from the previous one. Concerns over infection and the social distancing triggered by COVID-19 have changed people’s behavior and had significant impacts on their emotions, lifestyles, and social relationships [1,2]. When a novel infectious disease, such as COVID-19, spreads, the most common action taken by people is to immediately check the latest news related to the disease and obtain information [3]. Health information online is searched for, produced, and used through internet bulletin boards, communities, and search engines [4,5]. Internet use has both positive and negative impacts. Although the Internet provides opportunities for social interaction and belonging to individuals in isolation [6], at the same time, it can lead to an increase in negative emotions, such as anger, and serious social problems, such as addiction [7,8].

For people with disabilities (PWD), the development of high-speed Internet and social media environments such as Facebook and Twitter created a new chapter in which they can be active in all areas of life on an equal footing with the general public. PWD can talk or interact with a large number of people without time and space limitations, and work in various jobs by means of information technology [9,10]. It is also reported that informatization has a great contribution towards increasing the life satisfaction of PWD [11,12].

Despite the positive roles and effects of informatization, PWD, in particular, have long been classified as an information-vulnerable group. They are vulnerable in terms of access to information and communication technology and capability of using it, and are relatively less able to take advantage of the intelligent information society [13]. In other words, PWD may face information inequality and a gap depending on the accessibility of information, the quality of utilization, and the speed of access [14]. In an information society where information is an asset, the ability to use digital devices is a necessity for acquiring capital [15].

In South Korea, the Digital Divide Survey Report has been prepared every year since 2003. It reported the Internet use rate of the disabled to be lower than that of the non-disabled population annually up until the 2019 report, which was based on data surveyed before the onset of COVID-19. As non-face-to-face services spread owing to the COVID-19 situation, the problem of the digital divide for the information-vulnerable class emerged, and this expanded into social problems such as educational inequality and alienation from everyday social life [16,17]. Given the growing dependence on digital technologies such as artificial intelligence (AI), the Internet of Things, and big data in our daily lives [18,19], it is time to closely examine the impact of COVID-19 on information-vulnerable classes, such as the elderly and PWD, in the changing media environment. However, there are no empirical studies in this regard focused on PWD. Therefore, this study aims to investigate the impact of COVID-19 on Internet use of PWD. The specific research questions of this study are as follows.

First, what is the impact of COVID-19 on the PC and mobile device usage of PWD? Second, what is the impact of COVID-19 on the use of digital services by PWD? Third, how is the awareness and experience of COVID-19-related non-face-to-face services of PWD? Fourth, how useful are COVID-19-related non-face-to-face services for PWD?

## 2. Materials and Methods

### 2.1. Data

This study conducted a secondary data analysis using raw data targeting the general public and PWD, from the National Information Society Agency (NIA)’s ‘2020 Digital Divide Survey’ in South Korea. Since 2002, NIA has been collecting data every year to investigate the information gap for the vulnerable. The survey population was general citizens aged 7 years or above residing in households across South Korea as of 1 August 2020. The types of disabilities considered were physical disability, brain lesion, visual impairment, hearing impairment, and language impairment. The target group of the non-disabled population comprised citizens aged 7 years or older, including 2300 samples of the elderly, i.e., aged 55 years or older. The target group for disability comprised registered PWD aged 7 years or older nationwide, and the criteria for each type of disability were used. The sample size of the non-disabled population was 7000, and the number of PWD was 2200. In the case of the general public, the sample was extracted using the stratification probability proportional sampling method by place of residence, and in the case of PWD, the sample was extracted using proportional allocation sampling by gender, age, type of disability, and place of residence. The general characteristics of the samples are shown in Table 1. Data for experience on non-face-to-face services were restricted to people who were aware of those services and usefulness of service was restricted to people who used the services.

By type of disability, there were 1402 persons with physical impairment, 266 with brain damage, 273 with visual impairment, 208 with hearing impairment, and 51 with language impairment. By age group, those in their 60s or older accounted for the majority (28.6%), followed by those in their 50s (20.9%). An education level of high school graduation was achieved by 52.4%, accounting for more than half of the sample, and the residential area was mostly urban (93.4%). The proportion of PWD owning a PC was lower than that of the general public. The percentage of PWD possessing smart devices was higher than that of the general public.

### 2.2. Measures

#### 2.2.1. Changes in PC and Smart Device Usage

The changes in PC and smart device usage were measured by the following question: How has your PC and smart device internet usage changed due to COVID-19? This question was answered on five-point scales for PCs and smart devices, where 1 point corresponded to a very large decrease and 5 points corresponded to a very large increase.

Usage changes were measured by the following categories: search, email, and content services; social networking services (SNS); life support services; information production and sharing; degree of networking; degree of social participation; and degree of online economic activities. Each category was rated on a five-point scale.

#### 2.2.2. Awareness, Experience, and Usefulness of COVID-19-Related Non-Face-to-Face Services

The questionnaire consisted of four items: internet/mobile services such as application for government subsidies related to COVID-19, internet/mobile information services, internet/mobile delivery services, and internet/mobile subscription services. Awareness was measured by ‘Know’ or ‘Don’t Know’. Experience was measured by ‘Yes’ or ‘No’. The usefulness was rated on a four-point scale, with 1 being not useful at all and 4 being very useful.

#### 2.2.3. Statistical Analysis

A frequency analysis was conducted to measure Internet usage changes on PCs and smart devices using SPSS 25.0 (Armonk, NY, USA). Differences in Internet usage changes and awareness and experience of COVID-19-related non-face-to-face services by type of disabilities were assessed by χ^2^.

The differences in usage changes such as search, email, and content services; social networking services (SNS); life support services; information production and sharing; degree of networking, degree of social participation; and degree of economic activities were analyzed by ANOVA. The usefulness of COVID-19-related non-face-to-face services was also analyzed by ANOVA. Bonferroni post-hoc analysis was conducted.

Cohen’s d was calculated for the results of the COVID-19 effect on Internet usage changes on PC and smart devices. An effect size below 0.2 was interpreted as small, an effect size from 0.2 to 0.7 as medium, and an effect size above 0.7 as large.

## 3. Results

### 3.1. Changes in Internet Usage on PC and Smart Devices Due to COVID-19

Table 2 shows the effect of COVID-19 on Internet usage changes on PC and smart devices for the non-disabled population and PWD. Internet usage changes on PC for non-disabled population significantly increased compared with those for people with physical disabilities, brain lesion, visual impairment, and hearing impairment (*p* < 0.05). Internet usage changes on smart devices for the non-disabled population significantly increased relative to those for people with physical disabilities, brain lesion, visual impairment, and hearing impairment (*p* < 0.05). Internet usage changes for PWD were significantly higher than for people with brain lesion and visual impairment in smart devices (*p* < 0.05). The effect size (d) was from 0.1 to 0.5.

### 3.2. Changes in Internet Service Usage Due to COVID-19

Table 3 shows the effect of COVID-19 on Internet service usage changes in the non-disabled population and PWD. Internet service usage changes in all categories except social participation for the non-disabled population increased significantly more than for people with physical disabilities, brain lesion, visual impairment, and hearing impairment (*p* < 0.05). Regarding social participation service, the non-disabled population showed a significantly greater increase in usage changes than people with physical disabilities (*p* < 0.05).

### 3.3. Awareness and Experience of COVID-19-Related Non-Face-to-Face Services

Table 4 shows the awareness and experience of COVID-19-related non-face-to-face services. People with brain lesion and visual impairment had relatively low awareness of the services. Regarding non-face-to-face information service, people with physical disabilities and language impairment had relatively high awareness. Regarding COVID-19-related non-face-to-face delivery service, people with physical disabilities, brain lesion, visual impairment, and hearing impairment had relatively low awareness. Regarding COVID-19-related non-face-to-face subscription service, people with disabilities had low awareness.

PWD had relatively low experience of COVID-19-related non-face-to-face services. Regarding non-face-to-face information service and subscription, people with physical disabilities, brain lesion, visual impairment, and hearing impairment had relatively low experience. People with hearing impairment in particular showed the lowest experience of subscription among groups.

### 3.4. Usefulness of COVID-19-Related Non-Face-to-Face Services

Table 5 shows the usefulness of COVID-19-related non-face-to-face services. The usefulness of application service was significantly lower for PWD than for the non-disabled population. The usefulness of information service was significantly lower for people with hearing impairment than for the non-disabled population. The usefulness of delivery service was significantly lower for people with visual impairment than for the non-disabled population and those with physical disabilities. The usefulness of delivery service was significantly lower for people with physical disabilities than for the non-disabled population.

## 4. Discussion

This study analyzed the effect of COVID-19 on the change in Internet usage among PWD. The 2020 Digital Divide Survey data were utilized for the analysis. The major findings are discussed below.

First, COVID-19 had a significant effect on Internet usage changes on PC and smart devices for the non-disabled population and PWD. In particular, an increase in smart device usage was evident. However, the increase in Internet usage in the non-disabled population was significantly higher than in PWD, except in people with language impairment.

The digital divide can be considered as one possible reason for these results. Previous research findings on the digital divide for PWD have reported that PWD have a higher probability of experiencing the digital divide [20,21]. This may have been reflected in Internet usage changes. Another reason might be the age of PWD surveyed. The ratio of older people in PWD was larger than in the non-disabled population in this study, and many studies have reported lower Internet usage and competence in older people than in younger people [22,23].

Second, in terms of Internet usage for social participation, it was found that there was no significant difference between the general population and PWD except for people with physical disabilities. The COVID-19 pandemic has increased the importance of hearing the voices of PWD through social and digital media [24]. A study on social media usage by PWD during the COVID-19 pandemic analyzed data from 1374 people and showed that PWD outperformed people without disabilities in social media usage, staying up to date on COVID-19-related information, sharing information, and participating in various interactions about it [25].

Third, PWD showed relatively lower awareness and experience of COVID-19-related non-face-to-face services than the non-disabled population, but this difference depended on the type of disability. Among PWD groups, people with brain lesion and visual impairment had relatively lower awareness than people with other types of disabilities. People with hearing impairment showed the lowest experience of subscription among groups. This result shows that PWD is a fairly heterogeneous group, and this finding is consistent with the results of previous studies that compared and reported digital competence according to each type of disability [26]. The implication of this is that it is necessary to develop flexible and accessible technologies to adapt to times of crisis and provide services to individuals with diverse needs, as well as policies that can accommodate their unique needs according to types of disabilities.

Fourth, regarding the usefulness of non-face-to-face services related to COVID-19, PWD reported significantly lower usefulness than the non-disabled population. In particular, people with hearing impairment reported lower usefulness of information service, and people with visual impairment and physical disabilities reported lower usefulness of delivery service. Gleason and Valencia [27] analyzed Twitter data during the pandemic and found that, while essential retailers extended shopping hours and pick-up and delivery services to high-risk customers, PWD still lacked the necessary access to goods and services.

This study has several limitations. First, although this study analyzed systematically expressed large-scale sample group data, it was not possible to present an in-depth analysis of the content because the survey consisted of fragmentary questions. In future research, it is necessary to conduct more in-depth research on how the pandemic has affected the Internet life of PWD. Second, PWD are a heterogeneous group. Although this study analyzed how the pandemic affected people with different types of disabilities, future research is needed to investigate whether there were factors that affected the differences within people with each type of disability. The third limitation was related to limited information about comorbid conditions of PWD, which might have an effect on the Internet usage of PWD. Fourth, the ratio of elderly people in PWD was larger than in the non-disabled people. Although a systematic sample was used in analyzing data in this study, Internet usage changes due to COVID-19 might be affected by age. Further studies are recommended to verify the age effect.

## 5. Conclusions

The ongoing COVID-19 pandemic has forced people around the world to rely on residential Internet connections for work, education, social activities, and entertainment, and PWD are no exception to this. This study, which analyzed survey results during the COVID-19 pandemic, showed that the pandemic led to an increase in Internet usage for both the non-disabled population and PWD. However, the data analysis by group showed that the increase in usage differed according to the type of service and the type of disability. Even PWD who were aware of the non-face-to-face services related to the pandemic found those services less useful than the non-disabled population. Hence, it is suggested that there is a necessity to develop technologies in this regard that address the diverse needs of people with different types of disabilities. Moreover, relevant policies that fulfill the unique needs of different vulnerable groups need to be formed and implemented.

## Figures and Tables

**Table 1 ijerph-19-07813-t001:** General characteristics.

Variables	Non-Disabled Population	People with Disabilities					Total	χ^2^
		Physical Disabilities	Brain Lesion	Visual Impairment	Hearing Impairment	Language Impairment		
Sex								
Male	3505 (50.1)	972 (69.3)	181 (68.0)	188 (68.9)	132 (63.5)	28 (54.9)	5006 (54.4)	228.74 **
Female	3495 (49.9)	430 (30.7)	85 (32.0)	85 (31.1)	76 (36.5)	23 (45.1)	4194 (45.6)	
Age								
Under 20	871 (12.4)	3.00 (0.2)	32 (12.0)	0 (0.0)	10 (4.8)	5 (9.8)	921 (10.0)	915.74 **
20–29	995 (14.2)	26 (1.9)	9 (3.4)	8 (2.9)	5 (2.4)	0 (0.0)	1043 (11.3)	
30–39	1030 (14.7)	80 (5.7)	17 (6.4)	21 (7.7)	9 (4.3)	3 (5.9)	1160 (12.6)	
40–49	1183 (16.9)	224 (16.0)	27 (10.2)	54 (19.8)	26 (12.5)	7 (13.7)	1521 (16.5)	
50–59	1229 (17.6)	477 (34.0)	69 (25.9)	83 (30.4)	54 (26.0)	13 (25.5)	1925 (20.9)	
Above 60	1692 (24.2)	592 (42.2)	112 (42.0)	107 (39.2)	104 (50.0)	23 (45.1)	2630 (28.6)	
Education level								
Above UG	2062 (29.5)	376 (26.8)	72 (27.1)		50 (26.4)	14 (240)	2646 (28.8)	137.11 **
High school	3461 (49.9)	879 (62.7)	160 (60.2)	162 (59.3)	126 (60.6)	30 (58.8)	4818 (52.4)	
Middle school	1466 (20.9)	146 (10.4)	33 (12.4)	39 (14.3)	32 15.4)	7 (13.7)	1723 (18.7)	
Below ES	11 (0.2)	1 (0.1)	1 (0.4)	0 (0.0)	0 (0.0)	0 (0.0)	13 (0.1)	
Residential area								
City	6604 (94.3)	1266 (90.3)	229 (86.1)	253 (92.7)	189 (90.9)	49 (96.1)	8590 (93.4)	57.75 **
Urban	396 (5.7)	136 (9.7)	37 (13.9)	20 (7.3)	19 (9.1)	2 (3.9)	610 (6.6)	
PC owner								
Yes	5826 (83.2)	1009 (72.0)	137 (51.5)	159 (58.2)	120 (57.7)	23 (45.1)	7274 (79.1)	402.51 **
No	1174 (16.8)	393 (28.0)	129 (48.5)	114 (41.8)	88 (42.3)	28 (54.9)	1926 (20.9)	
Smart device owner								
Yes	6746 (96.4)	1385 (98.8)	265 (99.6)	271 (99.3)	206 (99.0)	49 (96.1)	8922 (97.0)	
No	254 (3.6)	17 (1.2)	1 (0.4)	2 (0.7)	2 (1.0)	2 (3.9)	278 (3.0)	

Note. ** *p* < 0.01; UG = undergraduate; ES = elementary school.

**Table 2 ijerph-19-07813-t002:** COVID-19 effect on Internet usage changes on PC and smart devices.

Categories	Non-Disabled Populations		Physical Disabilities		Brain Lesion		Visual Impairment		Hearing Impairment		Language Impairment		*F*
	M	SD	M (d)	SD	M (d)	SD	M (d)	SD	M (d)	SD	M (d)	SD	
PC	3.21 ^a^	0.65	3.14 ^b^ (0.1)	0.57	3.08 ^b^ (0.2)	0.48	3.07 ^b^ (0.2)	0.43	3.07 ^b^ (0.2)	0.53	3.06 ^ab^ (0.2)	0.65	8.12 **
Smart device	3.69 ^a^	0.67	3.56 ^b^ (0.2)	0.61	3.40 ^c^ (0.4)	0.56	3.38 ^c^ (0.5)	0.63	3.54 ^b^ (0.2)	0.60	3.65 ^abc^ (0.1)	0.59	27.22 **

Note. ** *p* < 0.01; subscripts such as ^a^, ^b^, and ^c^ provide expressions of post-hoc analysis results; means sharing the same subscripts are not significantly different at the *p* < 0.05 level through post-hoc analysis; d = Cohen’s d.

**Table 3 ijerph-19-07813-t003:** COVID-19 effect on Internet service usage changes.

Categories	Non-Disabled Populations		Physical Disabilities		Brain Lesion		Visual Impairment		Hearing Impairment		Language Impairment		F
	M	SD	M	SD	M	SD	M	SD	M	SD	M	SD	
Search, email, and content services	3.65 ^a^	0.74	3.41 ^b^	0.64	3.32 ^b^	0.65	3.28 ^b^	0.64	3.40 ^b^	0.65	3.57 ^a^	0.64	48.20 **
SNS	3.63 ^a^	0.73	3.35 ^b^	0.61	3.24 ^b^	0.62	3.26 ^b^	0.63	3.36 ^b^	0.68	3.45 ^a^	0.70	30.48 **
Life support service	3.50 ^a^	0.73	3.34 ^b^	0.68	3.20 ^c^	0.64	3.25 ^b^	0.66	3.22 ^b^	0.64	3.41 ^a^	0.70	14.10 **
Information production and sharing	3.27 ^a^	0.72	3.11 ^b^	0.58	3.03 ^b^	0.49	3.07 ^b^	0.54	3.00 ^b^	0.57	3.22 ^a^	0.61	12.41 **
Networking	3.28 ^a^	0.74	3.12 ^b^	0.60	3.04 ^b^	0.49	3.09 ^b^	0.62	3.12 ^b^	0.62	3.20 ^a^	0.60	10.74 **
Social participation	3.12 ^a^	0.71	3.06 ^b^	0.63	3.05 ^ab^	0.59	3.11 ^ab^	0.71	2.99 ^ab^	0.72	3.08 ^ab^	0.74	1.66 **
Online economic activities	3.59 ^a^	0.83	3.37 ^b^	0.65	3.18 ^b^	0.63	3.24 ^b^	0.61	3.28 ^b^	0.64	3.33 ^a^	0.62	26.16 **

Note. ** *p* < 0.01; subscripts such as ^a^, ^b^, and ^c^ provide expressions of post-hoc analysis results; means sharing the same subscripts are not significantly different at the *p* < 0.05 level through post-hoc analysis.

**Table 4 ijerph-19-07813-t004:** Awareness and experience of COVID-19-related non-face-to-face services.

Variables	Non-Disabled Population	People with Disabilities					Total	χ^2^
		Physical Disabilities	Brain Lesion	Visual Impairment	Hearing Impairment	Language impairment		
Awareness Application								
Know	6260 (89.4)	1181(84.2)	177 (66.5)	203 (74.4)	169 (81.3)	46 (90.2)	8036 (87.3)	192.95 **
Don’t know	740 (10.6)	221 (15.8)	89 (33.5)	70 (25.6)	39 (18.8)	5 (9.8)	740 (12.7)	
Information								
Know	4850 (69.3)	1127 (80.4)	176 (66.2)	188 (68.9)	162 (77.9)	44 (86.3)	6547 (71.2)	84.32 **
Don’t know	2150 (30.7)	275 (19.6)	90 (33.8)	85 (31.1)	46 (22.1)	7 (13.7)	2653 (28.8)	
Delivery								
Know	6015 (85.9)	1064 (75.9)	162 (609)	171 (62.6)	142 (68.3)	35 (68.6)	7589 (82.5)	295.76 **
Don’t know	985 (14.1)	338 (24.1)	104 (39.1)	102 (37.4)	66 (31.7)	16 (31.4)	1611 (17.5)	
Subscription								
Know	5067 (72.4)	679 (48.4)	96 (36.1)	97 (35.5)	84 (40.4)	20 (39.2)	6043 (65.7)	613.03 **
Don’t know	1933 (27.6)	723 (51.6)	170 (63.9)	176 (64.5)	124 (59.6)	31 (60.8)	3157 (34.3)	
Experience Application								
Use	4464 (71.3)	765 (64.8)	98 (55.4)	126 (62.1)	102 (60.4)	28 (60.9)	5583 (69.5)	52.33 **
Don’t use	1796 (28.7)	416 (35.2)	79 (44.6)	77 (37.9)	67 (39.6)	18 (39.1)	2453 (30.5)	
Information								
Use	4042 (83.3)	852 (75.6)	121 (68.8)	127 (67.6)	112 (69.1)	37 (84.1)	5291 (80.8)	92.13 **
Don’t use	808 (16.7)	275 (24.4)	55 (31.3)	61 (32.4)	50 (30.9)	7 (15.9)	1256 (19.2)	
Delivery								
Use	4561 (75.8)	737 (69.3)	96 (59.3)	98 (57.3)	83 (58.5)	20 (57.1)	5595 (73.7)	87.99 **
Don’t use	1454 (24.2)	327 (30.7)	66 (40.7)	73 (42.7)	59 (41.5)	15 (42.9)	1994 (26.3)	

Note. ** *p* < 0.01.

**Table 5 ijerph-19-07813-t005:** Usefulness of COVID-19-related non-face-to-face services.

Categories	Non-Disabled Populations		Physical Disabilities		Brain Lesion		Visual Impairment		Hearing Impairment		Language Impairment		*F*
	M	SD	M	SD	M	SD	M	SD	M	SD	M	SD	
Application	3.39 ^a^	0.53	3.30 ^b^	0.50	3.35 ^ab^	0.52	3.29 ^ab^	0.55	3.28 ^ab^	0.47	3.29 ^ab^	0.46	4.89 **
Information	3.35 ^a^	0.58	3.30 ^ab^	0.52	3.29 ^ab^	0.49	3.27 ^ab^	0.53	3.18 ^b^	0.47	3.27 ^ab^	0.45	3.90 **
Delivery	3.37 ^a^	0.57	3.26 ^b^	0.56	3.38 ^abc^	0.57	3.16 ^c^	0.55	3.31 ^abc^	0.52	3.55 ^abc^	0.51	7.82 **
Subscription	3.27	0.55	3.19	0.50	3.24	0.52	3.15	0.62	3.00	0.35	3.19	0.40	3.56 **

Note. ** *p* < 0.01; subscripts such as ^a^, ^b^, and ^c^ provide expressions of post-hoc analysis results; means sharing the same subscripts are not significantly different at the *p* < 0.05 level through post-hoc analysis.

## Data Availability

Data can be found at https://www.data.go.kr/data/15038422/fileData.do (accessed on 3 March 2022).

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
