# Peer review of "Effect of COVID-19 on Internet Usage of People with Disabilities: A Secondary Data Analysis"

_ijerph, 2022, doi:10.3390/ijerph19137813_

Round 1
Reviewer 1 Report
1. One of the most serious concerns for the findings of this study is confounding. As described in Table 1, people with disabilities are older than those without disabilities. This raises a possibility that “older people, but not people with disabilities, increased PC usage less than younger people, but not people without disabilities”. This should be addressed or discussed furthermore.
2. In Data paragraph in Materials and Methods section, sampling methods were described very shortly. This part should be expanded because it is vital to make sure to interpret the reliability of the findings.
3. In Table 2, 3, and 5, descriptions about a, b, c are not shown in the footnote. These should be written in each Table for readability.
Author Response
Thank you very much for giving me the opportunity to revise and resubmit this manuscript.
In the revised manuscript, I have highlighted in blue where I have made the major changes; please, note that I removed track changes that show all edits.
Comment #1: One of the most serious concerns for the findings of this study is confounding. As described in Table 1, people with disabilities are older than those without disabilities. This raises a possibility that “older people, but not people with disabilities, increased PC usage less than younger people, but not people without disabilities”. This should be addressed or discussed furthermore.
Response #2: I agreed to your comment. Related to your comment, discussion have been revised (line 200 ~ line 203; line 240 ~ line 243).
Comment #2: In Data paragraph in Materials and Methods section, sampling methods were described very shortly. This part should be expanded because it is vital to make sure to interpret the reliability of the findings.
Response #2: According to your comments, sampling methods was expanded (line 78 ~ line 86).
Comment #3: In Table 2, 3, and 5, descriptions about a, b, c are not shown in the footnote. These should be written in each Table for readability.
Response #3: The a, b, c was for express of post-hoc analysis. It has been inserted in each table (Table 2, 3, and 5).
Reviewer 2 Report
I want to thank you for the opportunity to review “Effect of COIVD-19 on Internet Usage of People with Disabilities: A Secondary Data Analysis”.
The study is using survey data to compare people with disability (areas of physical disabilities, brain lesion, visual impairment, hearing impairment, and language impairment) with people with out a disability on internet usage during the beginning of COVID (e.g., 2020). Four key areas of COIVD impact were looked at 1) PV vs mobile device, 2) use of digital services, 3) non-face-to-face services, 4) usefulness of non-face-to-face services.
It was great to see such a large sample size was collected. The analyses used in this study were appropriate. It would have been nice to have effect size computed for the difference report. As a 5 point liker scale was used, most changes were with in .5 of each other. Though significant, I wonder how meaningful of a difference this is.
A key finding in the study was that individuals with out a disability did indicate that they used the internet more during the first part of COIVD then people with a disability. However, the survey results were preserved change sores and not actual usage amounts. It may be that people with disability were using the internet more before the start of COIVD and therefor people with disabilities have caught up to them. It is critical to the argument that the authors address this issue.
The finding that different groups of disabilities had different effects on awareness of services is so important. Knowing which groups to target more will help with services provisions. However, the authors never address comorbid conditions. As we know many of these disabilities do not present in isolation. It would be good for the author to report if participates could indicate they had comorbid issues and if they could, how was this handled.
Author Response
Thank you very much for giving me the opportunity to revise and resubmit this manuscript.
In the revised manuscript, I have highlighted in blue where I have made the major changes; please, note that I removed track changes that show all edits.
Comment #1: It was great to see such a large sample size was collected. The analyses used in this study were appropriate. It would have been nice to have effect size computed for the difference report. As a 5 point liker scale was used, most changes were within .5 of each other. Though significant, I wonder how meaningful of a difference this is.
Response #1: In order to understand the meaning of the difference, the effect size was calculated and inserted in Table 2, the main result (Table 2).
Comment #2: A key finding in the study was that individuals without a disability did indicate that they used the internet more during the first part of COIVD then people with a disability. However, the survey results were preserved change sores and not actual usage amounts. It may be that people with disability were using the internet more before the start of COIVD and therefor people with disabilities have caught up to them. It is critical to the argument that the authors address this issue.
Response #2: In introduction, the statement about lower internet usage of people with a disability until COVID start was inserted (line 53 ~ line 56). Also, see the related background (line 45 ~ line 51).
Comment #3: The finding that different groups of disabilities had different effects on awareness of services is so important. Knowing which groups to target more will help with services provisions. However, the authors never address comorbid conditions. As we know many of these disabilities do not present in isolation. It would be good for the author to report if participates could indicate they had comorbid issues and if they could, how was this handled.
Response #3: Please understand that there is no information about comorbid conditions in the data. The limitations of this are described (line 238 ~ line 239).
Reviewer 3 Report
On the whole, this is a well-conducted and well-presented study. The aim is clear, it is easy reading, and the results are clearly laid out and discussed. It focuses on an area of the population often ignored during the grand sweep of the epidemic. Within the limitations described in the paper, it is a useful study.
No major changes are required.
A few small things:
· 1. I would like to see the Abstract laid out more formally, following the structure of the paper.
· 2. In addition, the Abstract should give a few more of the results.
· 3. The study speaks of the “National Information Society Agency (NIA)’s ‘2020 66 Digital Divide Survey’.”, but gives no location of this NIA. From the context described a little later and the location of the author, I am guessing that this is in South Korea, but it should be stated explicitly in order to remove any uncertainty.
Overall, a good paper. Nicely done.
Author Response
Thank you very much for giving me the opportunity to revise and resubmit this manuscript. I am grateful to you and the reviewer for insightful suggestions and comments. I have addressed all your suggested edits and comments. I hope that we have adequately strengthened our manuscript.
In the revised manuscript, I have highlighted in blue where I have made the major changes; please, note that I removed track changes that show all edits.
Comment #1: I would like to see the Abstract laid out more formally, following the structure of the paper.
Response #1: Abstract have been revised.
Comment #2: In addition, the Abstract should give a few more of the results.
Response #2: Abstract have been revised to insert more results (line 18 ~ line 20).
Comment #3: The study speaks of the “National Information Society Agency (NIA)’s ‘2020 66 Digital Divide Survey’.”, but gives no location of this NIA. From the context described a little later and the location of the author, I am guessing that this is in South Korea, but it should be stated explicitly in order to remove any uncertainty.
Response #3: Thank you for your helpful comment. Location has been inserted (line 74).
Round 2
Reviewer 1 Report
In Table 2, 3, and 5, please explain what a is, b is, and c is. Current description is not sufficient.
Author Response
Dear Reviewer
Thank you for your comments.
Comment #1: In Table 2, 3, and 5, please explain what a is, b is, and c is. Current description is not sufficient.
Response #2: According to your comments, I have revised the description. Please understand that the lower alphabet is for displaying the results of the post-hoc test in the table and does not have meaning for each.
Subscripts such as a, b, c provides expressions of post-hoc analysis results.; Means sharing same subscripts are not significantly different at the p < .05 level through post-hoc analysis.